# Identification of Protein–Phenol Adducts in Meat Proteins: A Molecular Probe Technology Study

**DOI:** 10.3390/foods12234225

**Published:** 2023-11-23

**Authors:** Fenhong Yang, Yingying Zhu, Xiaohan Li, Fengtao Xiang, Moru Deng, Wei Zhang, Wei Song, Hao Sun, Changbo Tang

**Affiliations:** 1State Key Laboratory of Meat Quality Control and Cultured Meat Development, Key Laboratory of Meat Processing, Ministry of Agriculture, Key Lab of Meat Processing and Quality Control, Ministry of Education, Jiangsu Collaborative Innovation Center of Meat Production and Processing, College of Food Science and Technology, Nanjing Agricultural University, Nanjing 210095, China; 2020108080@stu.njau.edu.cn (F.Y.); lixiaohanxh@stu.njau.edu.cn (X.L.); xiangfengtao@stu.njau.edu.cn (F.X.); d1670677025@163.com (M.D.); lww99706@163.com (W.Z.); wsong@njau.edu.cn (W.S.); haosun@njau.edu.cn (H.S.); 2Engineering Research Center of Magnetic Resonance Analysis Technology, Department of Food Nutrition and Test, Suzhou Vocational University, Suzhou 210005, China; yingy_zhu@163.com

**Keywords:** myofibrillar protein, caffeic acid, molecular probe technology, phenol–protein adducts

## Abstract

Plant polyphenols with a catechol structure can form covalent adducts with meat proteins, which affects the quality and processing of meat products. However, there is a lack of fast and effective methods of characterizing these adducts and understanding their mechanisms. This study aimed to investigate the covalent interaction between myofibrillar protein (MP) and caffeic acid (CA), a plant polyphenol with a catechol structure, using molecular probe technology. The CA-MP adducts were separated via sodium dodecyl sulfate–polyacrylamide gel electrophoresis (SDS-PAGE) and detected via Western blot and LC-MS/MS analyses. The Western blot analysis revealed that various specific adducts were successfully enriched and identified as bands around 220 kDa, 45 kDa, and two distinct bands between 95 and 130 kDa. Combined with the LC-MS/MS analysis, a total of 51 peptides were identified to be CA-adducted, corresponding to 31 proteins. More than 80% of the adducted peptides carried one adducted site, and the rest carried two adducted sites. The adducted sites were located on cysteine (C/Cys), histidine (H/His), arginine (R/Arg), lysine (K/Lys), proline (P/Pro), and N-terminal (N-Term) residues. Results showed that the covalent interaction of CA and MP was highly selective for the R side chain of amino acids. Moreover, the adducts were more likely to form via C-N bonding than C-S bonding. This study provides new insights into the covalent interaction of plant polyphenols and meat proteins, which has important implications for the rational use of plant polyphenols in the meat processing industry.

## 1. Introduction

Plant polyphenols are natural antioxidants that can protect meat products from oxidative deterioration by scavenging free radicals and inhibiting peroxidation [1,2]. However, plant polyphenols with a catechol structure, such as caffeic acid (CA), can also undergo oxidation to form quinones, which can react covalently with muscle proteins and form phenol–protein adducts which can affect the structure and functional properties of meat proteins [3,4], as well as the quality, nutritional, and digestive properties of meat products [5,6]. Therefore, it is important to understand the mechanism and characteristics of the covalent interaction between plant polyphenols and meat proteins and its impact on meat quality and processing.

Previous studies have shown that CA adduction can affect the structure and gel properties of myofibrillar protein (MP) [2,4]. However, due to the complexity of meat protein systems, there is still a lack of analytical methods of accurately characterizing phenol–protein adducts and their formation mechanisms. Most of the existing methods are based on indirect measurements, such as colorimetric assays, spectrophotometric assays, or gel electrophoresis, which have low sensitivity, specificity, and resolution [7,8]. Moreover, these methods cannot provide information on the identity, location, and number of the adducted sites on the protein molecules, which are crucial for understanding the molecular basis of the phenol–protein interaction [2,9]. Furthermore, these methods cannot quantify the extent of protein modification by plant polyphenols, which is essential for evaluating their effect on meat quality and processing [2,4]. Therefore, there is a need for a fast and effective method of characterizing phenol–protein adducts and their formation mechanisms at the molecular level.

Molecular probe technology is a powerful method for identifying the modification of proteins by natural active compounds (e.g., plant polyphenols) in complex systems. This method involves labeling the active compound with a molecular probe that can bind strongly to the target proteins and carry a detectable tag, such as fluorophores, isotopes, biotin, or bio-orthogonal groups [10,11]. The molecular probe–protein complexes can then be separated via intracellular imaging or SDS-PAGE in-gel fluorescence scanning for probes with isotopes or fluorophores [12] or via streptavidin magnetic beads for probes with biotin tags [13], which undergo a series of non-specific washing and specific elution operations to obtain the purified proteins or proteomes modified by the active compound [14]. The purified proteins or proteomes modified by the active compound can then be analyzed using methods such as Western blot and mass spectrometry to identify the peptide sequences and the adducted sites on the protein molecules [15] so that the mechanisms of interaction between the active compounds and proteins can be deeply explored. This method can provide information on the identity, location, and number of the adducted sites on the protein molecules, which are crucial for understanding the molecular basis of the active compound–protein interaction. Moreover, this method can also quantify the extent of protein modification by plant polyphenols by comparing the relative abundances of modified and unmodified peptides or proteins. In this strategy, multiple controls are required to exclude non-specific interacting proteins.

In this study, we synthesized a biotinylated derivative of caffeic acid (BioC) that can act as a molecular probe to identify protein targets responsible for caffeic acid’s covalent modification. Based on the molecular probe technology, we aimed to identify protein–phenol adducts in meat proteins. A method to enrich, isolate, and identify adducts was firstly established and its feasibility was validated; it was then applied to identify and preliminarily quantify specific phenol–protein adducts in the MP system. Combined with LC-MS/MS, the adducted peptides and amino acid sites were also comprehensively identified and quantified. Finally, the mechanism of the adduction reaction between CA and MP was explored based on the adducted sites and their relative abundance. This study provides new insights into the covalent interactions of plant polyphenols and meat proteins and their implications for meat quality and processing.

## 2. Materials and Methods

### 2.1. Materials

Caffeic acid and streptavidin magnetic beads were both obtained from Cell Signaling Technology (#5947; Beverly, MA, USA). Bovine serum albumin, biotin, trypsin, and polyvinylidene fluoride membranes were purchased from Sigma-Aldrich (Taufkirchen, Germany). An anti-biotin antibody (abs153474) was obtained from Absin Bioscience Inc. (Shanghai, China). Goat anti-rabbit IgG was purchased from Beijing Cowin Biotech Co., Ltd. (Beijing, China). Skimmed milk powder was obtained from Sinopharm Chemical Reagent Co., Ltd. (Shanghai, China). Other reagents were purchased from Genscript Biotech Co., Ltd. (Nanjing, China) unless otherwise specified.

### 2.2. Sample Preparation

The synthesis of the BioC was carried out according to a published experimental protocol [16]. Briefly, CA and N-Boc-ethylenediamine were dissolved in N,N-dimethylformamide, which was stirred at room temperature for 15 h in the presence of 1-(3-dimethylaminopropyl)-3-ethylcarbodiimide hydrochloride and 4-dimethylaminopyridine. When the reaction was then carried out with methyl tert-butyl ether, a yellow solid precipitated (intermediate 1). It was purified using a silica column (methylene chloride/methanol = 20:1, *v*/*v*) and recrystallized with methylene chloride/petroleum ether. Next, intermediate 1 was dissolved in dichloromethane, the treatment of which with methanolic hydrochloric acid solution for 15 h in an ice bath and subsequent concentration generated the desired intermediate 2. Finally, the amidation of intermediate 2 with D-(+)-biotin in a similar way as above afforded the BioC (Appendix A).

The extraction of the MP and the BioC solution’s oxidation treatment were carried out following the methods of Park [17] and Yang [16]. The MP was stirred and dissolved in phosphate buffer (PBS, 10 mM; 0.6 M NaCl; pH 6.2). For a concentration-dependent experiment, the BioC was dispersed into MP suspensions (1 mg/mL) to reach final concentrations of 0, 10, 50, 250, and 1250 μmol/g MP which were then incubated on a shaker (QB-210; Haimen Kylin-Bell Lab Instruments Co., Ltd., Nantong, China) at 4 °C for 12 h. For competition inhibition experiments, different concentrations of CA (0×, 1×, 4×, 16× of 250 μmol), the competitive inhibitor of the BioC, were firstly added and incubated at 4 °C for 12 h. The BioC was then added to obtain a final concentration of 250 µM with a final MP concentration of 1 mg/mL, and the samples were incubated for another 12 h at 4 °C.

### 2.3. Identification Method for Phenol–Protein Adducts

The separation of the MP-BioC adducts was performed according to Yang [17] with slight modifications. Streptavidin magnetic beads were gently mixed and moved (10 μL for each sample) into 1.5 mL microcentrifugation tubes, which were then placed on the magnetic rack so that they could be magnetically separated. The collected beads were washed with 1 mL of PBST buffer three times. The beads then could be mixed and incubated with 100 μL of the MP-BioC samples (1 mg/mL) at 4 °C for 40 min so that the BioC-derivatized proteins fully bound to the beads. After incubation, the beads were collected, and the supernatant was discarded. The beads were then subjected to non-specific protein washing with 1 mL of a washing buffer at least five times, with as much supernatant as possible removed and retained at the last moment. To release the MP-BioC adducts from the beads, the beads were resuspended in PBS buffer and mixed with 5 × Sample Buffer (125 mM Tris, 10% SDS, 0.25% BPB). The samples were boiled at 95 °C for 5 min, followed by cooling and centrifugation at 14,000 rpm for 1 min to collect the adducts in the supernatant, which were then subjected to SDS-PAGE or a Western blot analysis (see Section 2.6) in order to accomplish the identification and initial quantification of the polyphenol–protein adducts.

### 2.4. Validation of Method

Bovine serum albumin (BSA), as a model protein, has been widely studied in various fields due to the diversity of its physiological functions [18]. Consequently, to verify the validity of the above-mentioned method of identifying phenol–protein adducts, a BSA-BioC reaction system was established by using BSA as the substrate protein. First, the BioC was added to a BSA solution to obtain final concentrations of 50 and 250 µM with a final BSA concentration of 1 mg/mL. Meanwhile, the BSA solution was supplemented with the same concentration (50, 250 µM) of biotin or an equal volume of the solvent DMSO (1%, *v*/*v*) as a control. All the samples were incubated at 4 °C for 12 h. Then, the BSA-BioC adducts in the system were isolated and characterized using the method described in Section 2.3

### 2.5. Gel Electrophoresis

Samples were loaded onto SDS-PAGE equipment according to the method of Liu [19] with a 4–20% gradient gel. Small amounts of protein samples from Section 2.2 or Section 2.4 were first taken for SDS-PAGE to illustrate that there was no problem with the original protein samples prior to incubation with the streptavidin magnetic beads. Then, two 20 µL samples in Section 2.3 (i.e., enriched phenol–protein adducts) were subjected to SDS-PAGE. After staining with Coomassie brilliant blue, one gel was used for a Western blot analysis for the phenol–protein adducts’ identification and the other for a mass spectrometry analysis after in-gel digestion.

### 2.6. Western Blot Analysis

A Western blot analysis was used to detect bands of the BioC-adducted proteins according to Pi [20]. Briefly, after SDS-PAGE, an eBlot L1 fast wet protein transfer system (GenScript, Nanjing, China) was used to transfer the proteins to polyvinylidene fluoride (PVDF) membranes (0.45 mm) which were then blocked with 5% skimmed milk powder in Tris-buffered saline with Tween-20 (TBST, pH 7.4) for 1.5 h. After blocking, the membranes were washed with TBST three times (10 min/wash) and incubated with the anti-biotin antibody diluted 1:1000 in TBST (20 mM Tris-base, 137 mM NaCl and 5 mM KCl, 0.05% Tween-20) overnight at 4 °C, followed by incubation with goat anti-rabbit IgG (diluted 1:2000) for 2 h. A Chemiluminescence Imager (ImageQuant LAS 4000; General Electric Company, Boston, MA, USA) was used to capture the membranes.

### 2.7. LC-MS/MS Analysis

Samples were prepared using a filter-aided sample preparation (FASP) workflow, as described by Wiśniewski et al. [21]. The peptides digested by enzymes were subsequently cleaned, concentrated, and enriched using the stop-and-go-extraction tips (StageTips) method, as described by Rappsilber et al. [22]. The peptides were dissolved in a buffer containing 2% acetonitrile and 0.1% trifluoroacetic acid (TFA) for MS analysis. The elute was dried completely in a speedvac centrifuge. The peptides were suspended in a buffer containing 2% acetonitrile and 0.1% TFA ready for MS analysis. The peptides were separated via LC–MS/MS, using an Easy-nLC1200 (Thermo Fisher Scientific, Altham, GA, USA) coupled with an Orbitrap Exploris 480 Mass Spectrometer (Thermo Fisher Scientific, USA). A trap column (100 μm × 2 cm) and an analytical column (75 μm × 25 cm) packed with Reprosil-Pur C18 particles (Dr Maisch GmbH) were used to separate the peptides with mobile phase A (0.1% formic acid (FA) in water) and mobile phase B (0.1% FA in acetonitrile) in a 60 min gradient: 6 to 23% B in 38 min, 23 to 32% B in 12 min, and 32 to 80% B in 5 min, and then B was kept at 80% for 5 min. The flow rate was set at 300 nL/min. The Orbitrap Exploris 480 Mass Spectrometer was operated in a data-dependent acquisition mode with a spray voltage of 2 kV and a heated capillary temperature of 320 °C. MS1 data were collected at a high resolution of 60,000 with a mass range of 350 to 1500 *m*/*z*; the precursor intensity threshold was set at 5.0e^4^, and a maximum injection time of 20 ms was used. For each full MS scan, the top 10 most abundant precursor ions were selected for MS2 with an isolation window of 1.6 *m*/*z* and a higher-energy collision dissociation with a normalized collision energy of 30. MS2 spectra were collected at a resolution of 15,000. The target value was 5.0e^3^ with a maximum fill time of 20 ms and a dynamic exclusion time of 30 s.

### 2.8. Data Search

Raw data were processed and analyzed using Proteome Discoverer (PD) 2.4 SP1 (Thermo Fisher Scientific, USA) with default settings. PD was set up to search the database from NCBI. Human keratin and trypsin sequences were used as the contaminated database for proteomic searching. Trypsin was used as the digestion enzyme. Carbamidomethyl (C) was specified as the fixed modification. Oxidation (M), acetyl (Protein N-term), the BioC (C_21_H_28_N_4_O_5_S) (C, K, R, H, P, N-term + 446.16239 Da) and the possible degradation product of the BioC (C_9_H_8_O_4_) (C, K, R, H, P, N-term + 162.03169 Da) were specified as variable modifications. A q-value (FDR) cutoff for the precursor and protein level was applied at 1%. The processed data output was then exported to Excel for further usage.

### 2.9. Statistical Analysis

All the experiments were carried out in triplicate. Differences were evaluated using a one-way ANOVA, and means were compared using Duncan’s multiple comparison under the SAS system (version 8.2); *p* < 0.05 was considered statistically significant.

## 3. Results

### 3.1. Experimental Strategy

We developed a novel method for the enrichment and separation of phenol–protein adducts by using a BioC rather than CA in the reaction system to label proteins that can adduct with CA. The BioC is a biotinylated derivative of CA which can be recognized and captured by streptavidin magnetic beads with high specificity and affinity. However, to avoid the possible adsorption of other molecules on the surface of the streptavidin magnetic beads, non-specific washing steps are often required [23]. Therefore, four different washing buffers were tested to evaluate their washing effects on MP-BioC (0 μM, control) samples, including reagent 1, PBS buffer; reagent 2, PBS buffer (1% SDS); reagent 3, 25 mM Tris, 0.15 M NaCl, pH = 7.2; and reagent 4, 20 mM Tris-HCl, 2% SDS, pH 7.4. The non-specific washing step was repeated five times, and the supernatant was retained after magnetic separation (the fifth washing supernatant), which was then subjected to SDS-PAGE sample preparation together with proteins eluted from the streptavidin magnetic beads (the fifth elution solution).

As shown in Figure 1a, there was no residual protein in the supernatant after five non-specific washings regardless of which kind of washing buffer was used, indicating that five non-specific washing steps were sufficient. At the same time, proteins remaining in the fifth eluent were observed. It was also found that there were almost no protein bands in the lane corresponding to reagent 4, indicating that it effectively removed a large number of non-specific proteins adsorbed on the surface of the streptavidin magnetic beads after washing. Therefore, the reagent 4, which had the best effect, was selected as the washing buffer. In the follow-up tests, it was found that only a few specific phenol–protein adducts were identified in the experimental group compared with the control group (as shown in Figure 1b). We speculated that reagent 4, containing 2% SDS, may cause the loss of specific adducts while washing away non-specific proteins. To achieve a comprehensive identification of the formation mechanism of phenol–protein adducts, reagent 3 (25 mM Tris, 0.15 M NaCl, pH = 7.2) was finally selected to carry out the next experiment, and the identification of specific phenol–protein adducts was achieved based on both SDS-PAGE and a Western blot analysis.

### 3.2. Efficacy of the Identification Method of Phenol–Protein Adducts

Figure 2a shows a clear protein band around 66 kDa for BSA, which is consistent with its molecular mass [24]. Protein bands in all lanes are well defined, indicating that the protein samples were not degraded before incubation with streptavidin magnetic beads. Compared to the control group (BSA, BSA-DMSO, and BSA-Biotin), the band intensity at 66 kDa decreases and the bands near 95 kDa and above the separating gel increase as the BioC concentration increases. It is speculated that the BSA-BioC adducts were formed by a covalent reaction between the BioC and BSA which was proportional to the concentration. A series of control groups in the experiment could eliminate the interference of natural biotin-derived proteins in the samples. Previous studies have reported that phenolic compounds such as chlorogenic acid, catechin, and epicatechin can form covalent conjugates with BSA [25,26].

Figure 2b shows the immunoblotting result of BSA-BioC adducts enriched and isolated by the biotin–streptavidin system. No bands were labeled in any of the control groups, indicating that the streptavidin magnetic beads did not bind any non-specific proteins after washing. In contrast, the experimental groups showed a specific band between 66 and 95 kDa, and its intensity increased significantly with an increase in the BioC concentration. This confirmed that the BSA-BioC adducts were successfully enriched and isolated via the method described in Section 2.3, which proved to be effective and reliable. It is worth noting that increases in the protein band intensity of two bands were observed in the BSA-BioC samples in Figure 2a, but only one band appears in the Western blot result, which could be attributed to the S-S-mediated cross-linking of BSA rather than the covalent interaction between the BioC and BSA [27].

### 3.3. Identification of BioC-MP Adducts

#### 3.3.1. Concentration-Dependent Experiment

Figure 3a shows a typical electropherogram of the MP lane in which the main proteins are myosin heavy chain (MHC) around 220 kDa and Actin around 45 kDa [28,29]. When the BioC concentration increased, the MHC and Actin bands became weaker and moved to a higher molecular mass. As the concentration of the BioC increased, the MHC and Actin bands gradually weakened and shifted to a higher molecular mass. This suggests that MP-BioC adducts were generated in the system. Tang et al. [30] also found more than 20 proteins modified by rosmarinic acid in MP family proteins, including myosin heavy chain protein isoforms (MyHC1, MyHC2a, MyHC2x, and MyHC2b) and Actin. Since the feasibility of the method of identifying phenol–protein adducts was verified in Section 3.2, we carried out an enrichment of the adducts in the MP-BioC system following the same method and identified the MP-BioC adducts subsequently via a Western blot.

As shown in Figure 3b, the molecular probe, the BioC, marked out proteins at 220 kDa, 45 kDa, and 95–130 kDa as MP-BioC adducts, which depended on the concentration. Considering the low molecular mass of the BioC (MW = 448.185 Da), the position of the protein bands could hardly have shifted after the BioC modifications. Therefore, it was presumed that the protein bands around 220 kDa and 45 kDa were MHC-BioC adducts and Actin-BioC adducts respectively, and other the labeled bands were BioC covalent conjugates of the corresponding proteins.

In Figure 4, a preliminary quantitative analysis of MP-BioC adduct bands at 220 kDa, 95–130 kDa, and 45 kDa shows that the relative intensity of these adduct bands at a concentration of 1250 µM was significantly higher than that at other concentrations (*p* < 0.05). Specifically, the Actin-BioC and band-2-BioC adducts were firstly observed at a low BioC concentration (50 µM). MHC-BioC and band-1-BioC were observed at a BioC concentration of 250 µM. With an increasing BioC concentration, the intensity of all of adducts was significantly increased (*p* < 0.05). The results illustrate that the formation of MP-BioC adducts has concentration-dependent characteristics.

#### 3.3.2. Competition Suppression Experiment

The characteristic protein bands identified in the concentration-dependent experiments decreased gradually when we added different concentrations of highly active ligands. This was an effective way to confirm specific protein bands. In our study, we further tested the covalent interaction between molecule probes and highly active ligands that were added simultaneously to establish a competition control which could prevent the molecule probes from binding to target proteins via competition from highly active ligands. The addition of different concentrations of highly active ligands may lead to a gradual fading of the characteristic protein bands identified in concentration-dependent experiments, which was an effective method of determining specific protein bands. In this study, to further validate the covalent interaction between the BioC and MP, we allowed the BioC or CA to compete for binding to MP with or without an excess of the competitive inhibitor CA, and the BioC concentration was kept at 250 µM. Figure 5a shows that the MP-BioC samples in the competitive suppression experiment were normal. Figure 5b and Figure 6 show that the MP-BioC adducts decreased significantly (*p* < 0.05) when the CA concentration increased. When the CA concentration was 16 times higher than the BioC concentration (4 × 10^3^ µmol/g MP), the binding of the BioC to MP was almost completely blocked.

The protein bands at 220 kDa and 45 kDa, as well as two strong protein bands between 95 and 130 kDa, depended on the concentration and could be stopped by the competitive inhibitor CA. This indicated that all these protein bands were specific MP-BioC adducts. More importantly, the results showed that the BioC and MP interacted in the same way as the CA and MP. Since the BioC was modified from CA and had the same biological activity as CA (Appendix A), it was considered that adducts labeled by BioC such as MHC-BioC and Actin-BioC labeled were equivalent to adducts generated by the covalent interaction of CA and MP.

### 3.4. LC-MS/MS of BioC-MP Adducts

#### 3.4.1. Identification of Adducted Proteins and Peptides

To further explore the formation mechanism of MP-BioC adducts, the BioC-adducted proteins in the MP sample (1250 µM) were identified using an Orbitrap Exploris 480 mass spectrometer. After filtering data with a score > 10, a total of 31 BioC-adducted proteins were found, including myosin, actin, troponin, and tropomyosin (Table 1). Six BioC-modified myosin isoforms were identified near 220 kDa, which included myosin-4 isoform X1 (223.1 kDa), myosin heavy chain 2x (223 kDa), myosin-2 isoform X1 (223 kDa), myosin-8 isoform X1 (222.9 kDa), embryonic skeletal myosin heavy chain 3 partial (222.8 kDa), and myosin-7B isoform X5 (223.8 kDa). Among them, the first five proteins had a BioC modification at the R site and the common adducted peptide was FIR*IHFGTTGK (* denotes the adducted site). BioC modifications at the N-Term site were also identified. At the same time, skeletal myosin heavy chain 3 had BioC modifications at the H and N-Term sites (IEDMAMLTH*LNEPAVLYNLK) and myosin 7b isomer X5 at the K site (LLGSLDIDHSQYQFGHTK*). At around 45 kDa, skeletal alpha actin (42 kDa), actin aortic smooth muscle (42 kDa), cardiac α actin 1 (pdb 5NOL B Chain B Cardiac muscle alpha actin 1, 40.7 kDa), and actin-partial (27.4 kDa) were modified with BioC at the R-site (EIVR*DIK) and N-Term site (MQKEITALAPSTMK). Three BioC-adducted peptides were identified in cytoskeletal beta actin partial (44.8 kDa), two of which were the same as above, and the other was DLYANTVLSGGTTMYPGIADR, with the N-Term as the adducted site.

In the molecular mass range of 95–130 kDa, the BioC modification of alpha-actinin-3 isoform X2 (103.2 kDa), glycogen phosphorylase muscle form (97.2 kDa), and muscle glycogen phosphorylase partial (84 kDa, adducted peptide FSAYLER) occurred at the N-Term site (DGLALCALIHR, FSAYLER, and FSAYLER, respectively), and that of FH1/FH2 domain-containing protein 1 isoform X1 (129.2 kDa) was located at two P sites (SPFPPPPPPAAP*LP*PSAPDGLALPTK). In addition, the alpha2 chain of type I collagen (129.1 kDa) had two adducted sites P and R (GEVGPAGPNGFAGPAGAAGQP*GAKGER*GTK). Furthermore, lower-molecular-mass adducts were identified, such as C-U-editing enzyme APOBEC-2 (25.8 kDa), GTP-binding protein SAR1b (22.4 kDa), and troponin I (21.3 kDa). BioC modifications of them occurred at the C or N-Term sites, and the adducted peptides involved were TFLC*YVIEAQSK, ELNARPLEVFMC*SVLK, and MSADAMLK, respectively.

Myosin, an important component of MP, has a molecular structure rich in charged amino acids and cysteines [31], which are susceptible to be attacked by the reactive electrophilic intermediate o-benzoquinone, leading to a Michael addition reaction. Tang et al. [32] investigated the interaction of RosA and meat proteins under the Fenton oxidizing system and showed that a diversity of myosins were involved in RosA modifications at K, R, and C sites. The difference was that the CA in our study mainly interacted with myosin at the alkaline amino acid (R, K, H) and N-Term sites of myosin but did not involve the C site. Instead, modifications at the C site were identified in low-molecular-mass proteins such as C-U-editing enzyme APOBEC-2 and GTP-binding protein SAR1b. Nikolantonaki et al. [33] noted that there was an adducted formation between caffeic acid and 3-sulfanylhexan-1-ol. It was speculated that oxidation conditions affect the covalent interaction of phenols and proteins to some extent. Β-actin is a major component of cytoskeleton [34], and polyphenol modifications to it can have an effect on gel properties during meat processing [28]. Meanwhile, muscle glycogen phosphorylase is a key enzyme for glycogen degradation and affects the differentiation of skeletal muscle cells. Therefore, it is hypothesized that BioC modification may regulate energy metabolism.

#### 3.4.2. Identification of Adducted Amino Acid Sites

Altogether, 51 BioC-adducted peptides were identified among 31 MP-BioC-adducted proteins (Table 2). BioC adduction occurred at six different amino acid sites, including the C, H, R, K, P, and N-Term sites. Secondary mass spectra of peptides with the BioC adducted on the C, H, R, K, and P sites, respectively, are shown in Figure 7a–e.

Distribution of adducted sites: Figure 8a shows that 59 BioC-adducted sites were identified in the BioC-adducted peptides, including 2 sites at C, 2 sites at H, 12 sites at R, 3 sites at K, 8 sites at P, and 32 sites at the N-term. Among them, alkaline amino acids (K, R, and H) accounted for 28.81% of the adducted sites. The proportions of C, P, and N-Term were 3.39%, 13.56%, and 54.24%, respectively. The result indicates that the BioC mainly modified the N-Term, followed by the alkaline amino acids P and C. In Figure 8b, a total of 43 BioC-adducted peptides (84.31%) contained one adducted site, and the remaining 8 (15.69%) carried two adducted sites, which suggests that the BioC preferred to react with MP at a single site. This result was similar to that of Tang et al. [2], who reported that RosA had the highest frequency of single-site addition reactions, followed by two-site addition reactions, and a low frequency of three or more site addition reactions (*p* < 0.05).

In unoxidized phenolic compounds, hydroxyl groups are excellent hydrogen donors, which do not directly form irreversible covalent bonds with amino acids but allow for the formation of hydrogen bonds between phenols and carboxyl groups in proteins. Under alkaline conditions, hydroxyl groups are oxidized and converted into the active derivative quinone. The latter then undergoes irreversible covalent addition reactions with the sulfhydryl or amino groups of susceptible amino acid residues in proteins [35,36]. Some studies have found that the N-Term of protein is thought to be strongly related to the stability of protein structure [37]. Based on this, the high proportion of N-Term modifications might correspond to changes in protein structure. It has been reported that peptides containing alkaline amino acid are mostly located inside protein [2], so it could be inferred that the presence of steric hindrance made them less reactive with the BioC. It should also be noted that the oxidation level, type, and dose of phenolic compounds will also affect the extent of their interaction with proteins [38].

The sequence motif of adducted sites: The full-length sequence of the different adducted peptides was searched in the NCBI according to the gene number before selecting the adducted site and 10 amino acids on each side of it to form a polypeptide with 21 amino acids. The sequence logo (seqlogo) was drawn using Weblogo (Version 2.8.2) to further explore the sequence motif of the adducted sites. The height (%) of each letter in the seqlogo corresponds to the frequency of the occurrence of amino acid residues at that location, and that letters in each position are arranged in order of conservatism from the largest to smallest, which makes it easy to identify conserved sequences from the top letters. As shown in Figure 9, the most frequent amino acids at the BioC-adducted site (coordinate: 0) were R, P, K, C, and H, in decreasing order. About 85% of them were X-R/P/K-X motifs, and the rest were X-C/H-X motifs (X = any amino acid). This result suggests that R had the highest conservation. In addition, it was speculated that the sequence motif of the BioC-adducted peptide was DNSSAEGEFIRDIKGKLGKLA, which has implications for the future study of meat protein.

## 4. Conclusions

In this study, we developed and validated a novel method based on molecular probe technology to accurately identify phenol–protein adducts in meat protein systems, using BSA as a model protein. Our method involves the use of a BioC, a biotinylated derivative of CA which can covalently bind to MP in the same manner as CA. It was demonstrated that the BioC, a biotinylated derivative of CA, can interact with MP in the same way as CA, forming covalent bonds with specific amino acid residues. By using an LC-MS/MS analysis, 31 CA-adducted proteins, 51 CA-adducted peptides, and 6 kinds of CA-adducted sites were identified. Furthermore, it was revealed that CA prefers to react with the -NH_2_ group (H, R, K, P, and N-term adducted sites) rather than the -SH group (C adducted site) of MP, resulting in various phenol–protein adducts. Our method provides a powerful tool for the isolation and identification of phenol–protein adducts in meat protein systems as well as the comprehensive analysis of the types and numbers of adducted sites, which reveal the mechanism of the covalent binding of phenols to proteins.

Although phenolic compounds have many beneficial effects on the quality, safety, and functionality of meat and meat products, there are some potential drawbacks and risks that need to be considered. One of the major challenges of using phenolic compounds in the meat industry is the oxidation of phenolic compounds with the formation of quinones, which could affect the quality, safety, and functionality of meat and meat products. The oxidation of phenolic compounds and the formation of quinones are influenced by many factors, such as the type, concentration, source of phenolic compounds, pH, temperature, oxygen, metal ions, enzymes, and other food components of the meat system, and the processing and storage conditions of the meat products. In this study, covalent interactions between plant polyphenols and meat proteins were investigated. However, only low-molecular-weight polyphenols were used, which may differ from high-molecular-weight polyphenols, such as proanthocyanidins and more complex tannins, in their interaction patterns and biological effects. High-molecular-weight polyphenols might form more stable covalent complexes with proteins. A future investigation is therefore required to take into account the molecular characteristics of polyphenols and the environmental factors that may influence the interactions between polyphenols and proteins and their effects on meat quality.

## Figures and Tables

**Figure 1 foods-12-04225-f001:**
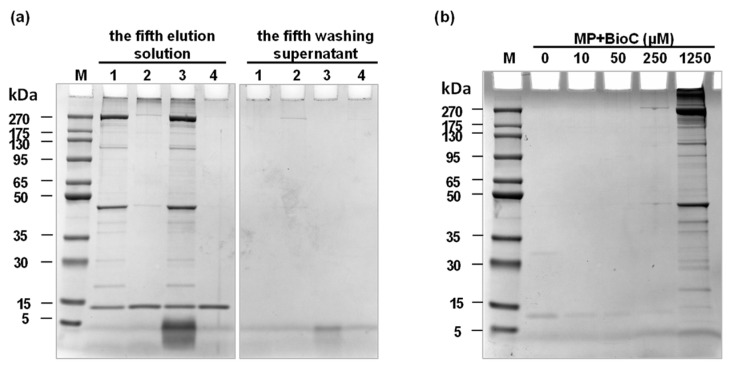
Non-specific washing using 4 kinds of washing buffers, reagent 1, 2, 3, and 4 (**a**); SDS-PAGE of MP-BioC adducts eluted from streptavidin magnetic beads after the fifth washing using washing reagent 4 (**b**).

**Figure 2 foods-12-04225-f002:**
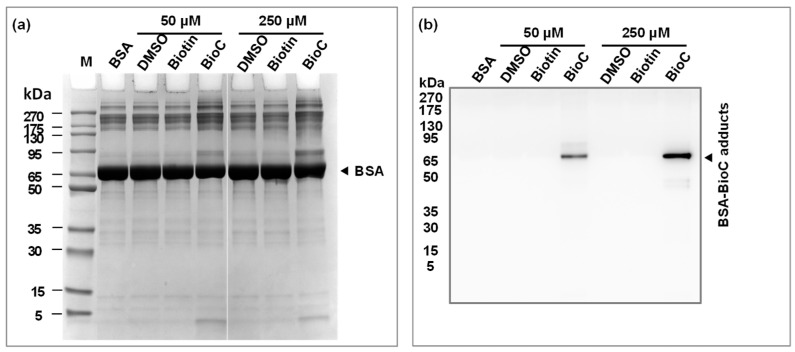
SDS-PAGE analysis of samples (**a**); Western blot analysis of BSA-BioC adducts (**b**).

**Figure 3 foods-12-04225-f003:**
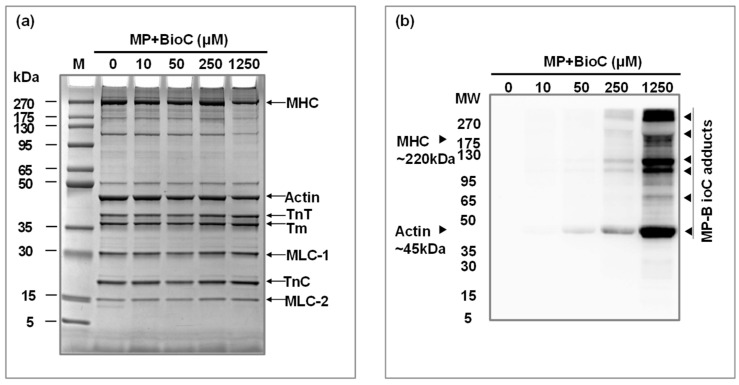
SDS-PAGE analysis of samples (**a**) and Western blot analysis of MP-BioC adducts (**b**) in the concentration-dependent experiment.

**Figure 4 foods-12-04225-f004:**
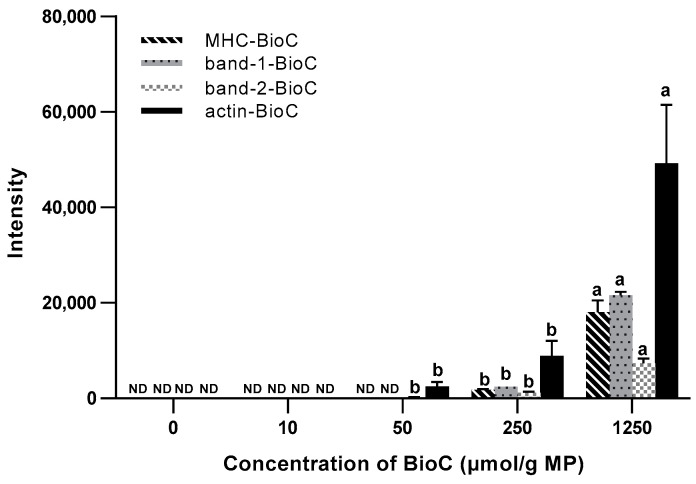
Types and intensity of MP-BioC adducts in the presence of different concentrations of BioC. Letters (a,b) indicate a significant difference (*p* < 0.05) in the relative intensity of MP-BioC adducts in groups with different concentrations of BioC; Band-1/2-BioC—the upper/lower one of two adducts between 95 and 130 kDa; ND—not detected.

**Figure 5 foods-12-04225-f005:**
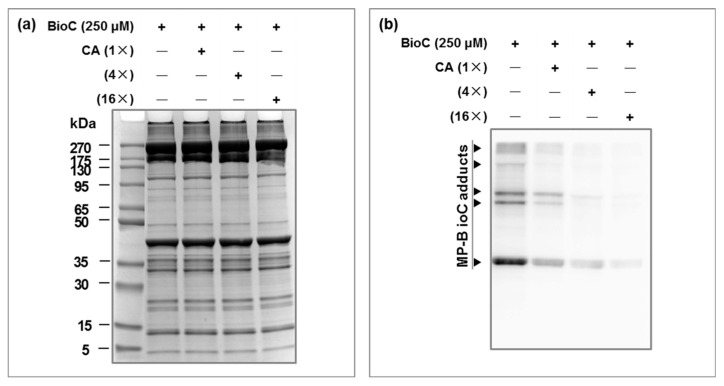
SDS-PAGE analysis of samples (**a**), and Western blot analysis of MP-BioC adducts (**b**) in the competition suppression experiment.

**Figure 6 foods-12-04225-f006:**
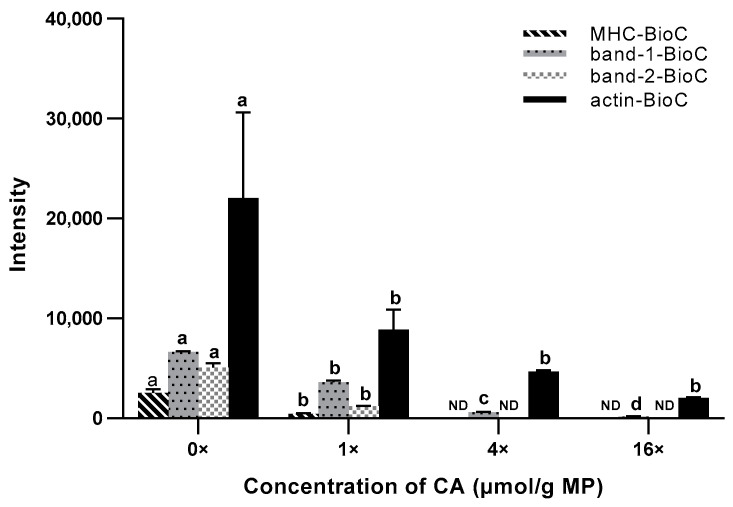
Types and intensity of MP-BioC adducts in the presence of different concentrations of CA. Letters (a–d) indicate a significant difference (*p* < 0.05) in the relative intensity of MP-BioC adducts with different concentrations of the competitive inhibitor CA (0×, 1×, 4×, 16× of 250 μmol); ND indicates not detected.

**Figure 7 foods-12-04225-f007:**
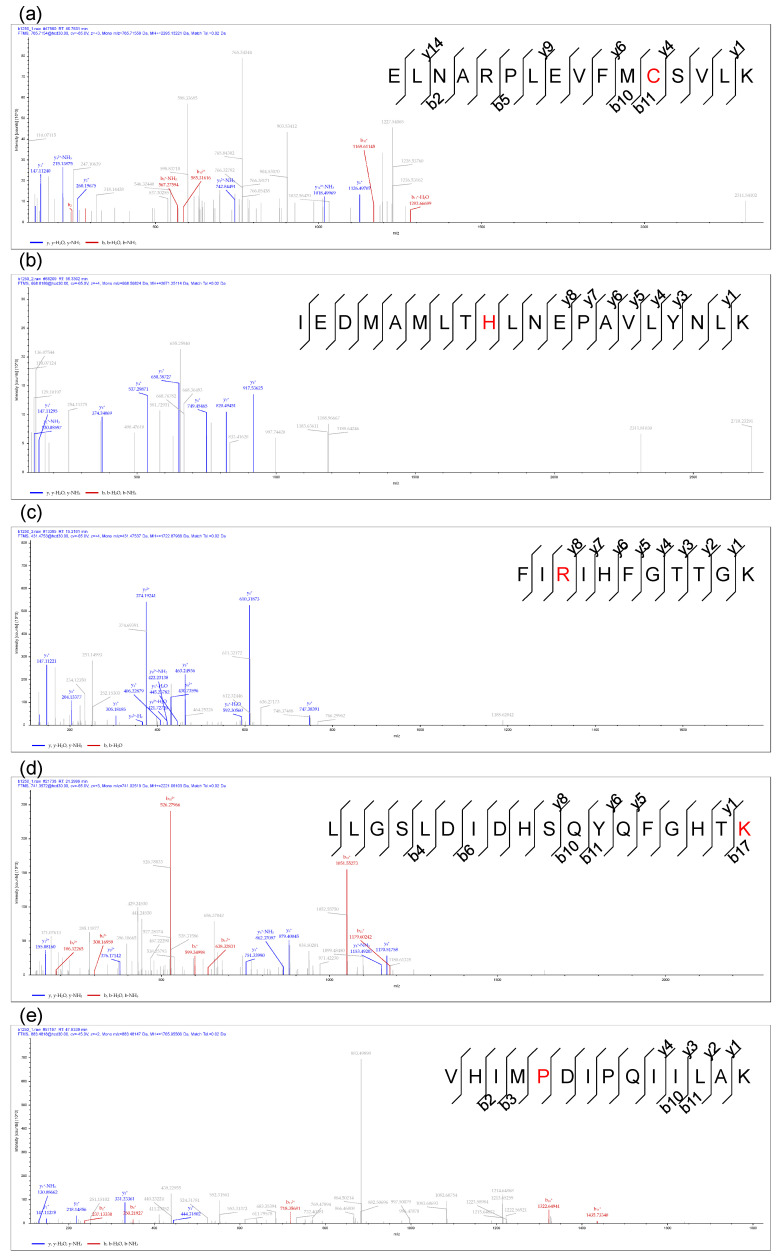
Secondary mass spectra of peptides with BioC adducted on different amino acid sites.

**Figure 8 foods-12-04225-f008:**
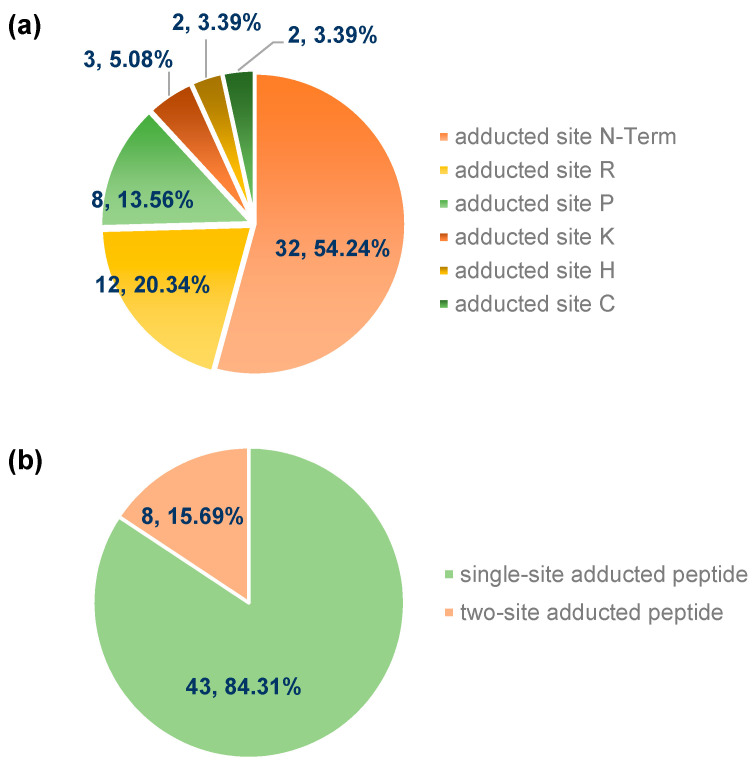
Distribution of different adducted sites (**a**); the number of adducted peptides carrying single or two sites (**b**).

**Figure 9 foods-12-04225-f009:**
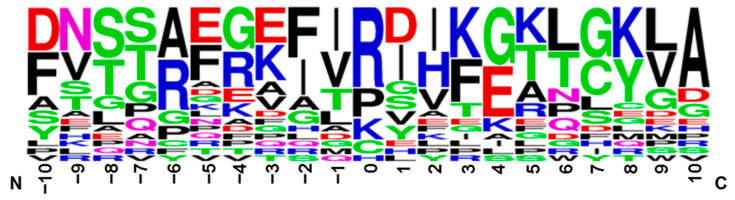
The seqlogo of the adducted peptides.

**Table 1 foods-12-04225-t001:** BioC-adducted proteins in MP.

Gene	Protein	Coverage (%)	MW (kD)	Calc. pI	Score
XP 020931560.1	LOW QUALITY PROTEIN titin	59	3816	6.27	17,031.96
XP 020921695.1	myosin-4 isoform X1	75	223.1	5.76	7827.33
BAA82146.1	myosin heavy chain 2x	73	223	5.77	6990.96
XP 020921876.1	myosin-2 isoform X1	72	223	5.81	5401.78
XP 020921928.1	myosin-8 isoform X1	50	222.9	5.8	4251.75
XP 020931125.1	LOW QUALITY PROTEIN nebulin	67	856.7	9.07	4104.53
AAC48692.1	skeletal alpha actin	76	42	5.39	2977.1
/	pdb 5NOL B Chain B Cardiac muscle alpha actin 1	77	40.7	5.57	2436.99
NP 001158122.1	actin aortic smooth muscle	76	42	5.39	2145.94
ART85712.1	embryonic skeletal myosin heavy chain 3 partial	28	222.8	5.86	1905.26
AAS55927.1	cytoskeletal beta actin partial	51	44.8	5.83	1595.71
XP 003122540.1	alpha-actinin-3 isoform X2	69	103.2	5.45	1384.5
ACD13863.1	actin partial	81	27.4	5.05	1298.08
XP 020946026.1	plectin isoform X4	34	532.3	6.09	799.32
JAA53758.1	Plectin partial	37	402.7	5.68	673.79
XP 013840744.1	myosin-7B isoform X5	6	223.8	6	505.58
XP 003122636.2	glycogen phosphorylase muscle form	58	97.2	7.11	370.28
ABF81978.1	muscle glycogen phosphorylase partial	63	84	6.52	370.18
NP 001123421.1	creatine kinase M-type	68	43	7.09	251
ACE80199.1	troponin I	47	21.3	8.91	200.06
XP 001928934.3	C- U-editing enzyme APOBEC-2	57	25.8	4.81	89.24
XP 020949729.1	FH1/FH2 domain-containing protein 1 isoform X1	13	129.2	6.32	35.59
XP 003356007.1	EH domain-containing protein 2	20	61.2	6.37	35.33
NP 001008689.1	GTP-binding protein SAR1b	34	22.4	6.11	32.56
BAX02569.1	alpha2 chain of type I collagen	9	129.1	9.07	31.72
NP 001231230.1	importin-5	12	123.6	4.94	29.83
XP 020922659.1	keratin type I cytoskeletal 14	7	51.6	5.16	27.27
NP 001258644.1	laminin subunit gamma-1 precursor	8	177.5	5.15	22.28
NP 001231936.1	eIF5-mimic protein 1	13	48.1	6.68	18.44
XP 013849669.2	LOW QUALITY PROTEIN neuroblast differentiation-associated protein AHNAK	12	654.3	6.05	13.91
XP 020939127.1	regulator of nonsense transcripts 1 isoform X1	6	125.5	6.7	13.08

**Table 2 foods-12-04225-t002:** BioC-adducted peptides and amino acid sites.

Gene	Protein	Adducted Peptide	Adducted Amino Acid Site
XP 020931560.1	LOW QUALITY PROTEIN titin	[K].NAAGVISK.[G]	[N-Term]
		[K].VDGTPEIR.[I]	[P5]
XP 020921695.1	myosin-4 isoform X1	[K].FIRIHFGTTGK.[L]	[R3]
		[K].QAFTQQIEELKR.[Q]	[N-Term]
		[R].VIQYFATIAVTGEK.[K]	[N-Term]
		[R].IEAQNKPFDAK.[T]	[N-Term]
BAA82146.1	myosin heavy chain 2x	[K].FIRIHFGTTGK.[L]	[R3]
		[K].QAFTQQIEELKR.[Q]	[N-Term]
		[R].VIQYFATIAVTGEK.[K]	[N-Term]
		[R].IEAQNKPFDAK.[T]	[N-Term]
XP 020921876.1	myosin-2 isoform X1	[K].FIRIHFGTTGK.[L]	[R3]
		[R].VIQYFATIAVTGEK.[K]	[N-Term]
XP 020921928.1	myosin-8 isoform X1	[K].FIRIHFGTTGK.[L]	[R3]
		[R].VIQYFATIAVTGEK.[K]	[N-Term]
		[R].IEAQNKPFDAK.[T]	[N-Term]
ART85712.1	embryonic skeletal myosin heavy chain 3 partial	[K].FIRIHFGTTGK.[L]	[R3]
		[K].QAFTQQIEELKR.[Q]	[N-Term]
		[R].IEDMAMLTHLNEPAVLYNLK.[D]	[H9]; [N-Term]
XP 003122540.1	alpha-actinin-3 isoform X2	[K].DGLALCALIHR.[H]	[N-Term]
XP 003122636.2	glycogen phosphorylase muscle form	[K].FSAYLER.[E]	[N-Term]
ABF81978.1	muscle glycogen phosphorylase partial	[K].FSAYLER.[E]	[N-Term]
NP 001123421.1	creatine kinase M-type	[R].SIKGYTLPPHCSR.[G]	[K3]; [N-Term]
ACE80199.1	troponin I	[R].MSADAMLK.[A]	[N-Term]
XP 001928934.3	C- U-editing enzyme APOBEC-2	[K].TFLCYVIEAQSK.[G]	[C4]; [N-Term]
XP 003356007.1	EH domain-containing protein 2	[R].TVTSALK.[E]	[N-Term]
NP 001008689.1	GTP-binding protein SAR1b	[K].ELNARPLEVFMCSVLK.[R]	1xBioC [C12]
NP 001231936.1	eIF5-mimic protein 1	[R].VQQSLGTR.[K]	[N-Term]
XP 020931125.1	LOW QUALITY PROTEIN nebulin	[R].KQLGHHVGAR.[N]	[H5]
		[K].VHIMPDIPQIILAK.[A]	[P5]
AAC48692.1	skeletal alpha actin	[R].EIVRDIK.[E]	[R4]
		[R].MQKEITALAPSTMK.[I]	[N-Term]
NP 001158122.1	actin aortic smooth muscle	[R].EIVRDIK.[E]	[R4]
		[R].MQKEITALAPSTMK.[I]	[N-Term]
/	pdb 5NOL B Chain B Cardiac muscle alpha actin 1	[R].EIVRDIK.[E]	[R4]
		[R].MQKEITALAPSTMK.[I]	[N-Term]
ACD13863.1	actin partial	[R].EIVRDIK.[E]	[R4]
		[R].MQKEITALAPSTMK.[I]	[N-Term]
AAS55927.1	cytoskeletal beta actin partial	[R].EIVRDIK.[E]	[R4]
		[R].MQKEITALAPSTMK.[I]	[N-Term]
		[K].DLYANTVLSGGTTMYPGIADR.[M]	[N-Term]
XP 020946026.1	plectin isoform X4	[R].SMVEEGTGLR.[L]	[N-Term]
JAA53758.1	plectin partial	[R].SMVEEGTGLR.[L]	[N-Term]
XP 013840744.1	myosin-7B isoform X5	[K].LLGSLDIDHSQYQFGHTK.[V]	[K]
XP 020949729.1	FH1/FH2 domain-containing protein 1 isoform X1	[K].SPFPPPPPPAAPLPPSAPDGLALPTK.[R]	[P12]; [P14]
BAX02569.1	alpha2 chain of type I collagen	[R].GEVGPAGPNGFAGPAGAAGQPGAKGERGTK.[G]	[P21]; [R27]
NP 001231230.1	importin-5	[K].FLFDSVSSQNMGLR.[E]	[N-Term]
XP 020922659.1	keratin type I cytoskeletal 14	[R].TKYETELNLR.[M]	[K2]; [N-Term]
NP 001258644.1	laminin subunit gamma-1 precursor	[R].KTLPSGCFNTPSIEKP.[-]	[P11]
XP 013849669.2	LOW QUALITY PROTEIN neuroblast differentiation-associated protein AHNAK	[K].AEGGGAEVQLPSLEGGLSMPDVK.[L]	[P11]; [N-Term]
		[K].GPGIDVKAPK.[M]	[P2]; [N-Term]
XP 020939127.1	regulator of nonsense transcripts 1 isoform X1	[R].MHPALSAFPSNIFYEGSLQNGVTAADRVK.[K]	[R27]

## Data Availability

The data used to support the findings of this study can be made available by the corresponding author upon request.

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
