# Peer review of "Identification of Protein–Phenol Adducts in Meat Proteins: A Molecular Probe Technology Study"

_foods, 2023, doi:10.3390/foods12234225_

Round 1

Reviewer 1 Report

Comments and Suggestions for Authors

In the paper were studied the interactions between the plant phenols and met proteins and their implication for meat quality and processing. The authors synthetized a biotinylated derivative of caffeic acid that was used as molecular probe to identify protein targets responsible for caffeic acid’s covalent modification. A plant polyphenol, caffeic acid (CA) was selected in the study for the investigation of the covalent interaction between the myofibrillar protein (MP) and phenols.

A biotinylated derivative of CA, which have similar biochemical reactions as CA was used. The adducts formed between CA and MP were separated using sodium dodecyl sulfate–polyacrylamide gel electrophoresis (SDS-PAGE) and detected by Western blot and LC-MS/MS analysis. The adducted peptides and amino-acid sites were identified and quantified using LCMS/MS method. The study is important due to the detailed information concerning the interaction between plant polyphenols and meat proteins. The results of the study are of interest for a rational use of plant polyphenols in the meat processing industry.

References to related papers and previous research are appropriate.

Conclusions are consistent with the objectives proposed by the research and with the results shown and discussed in the paper.

Observations:

1)The method of synthesis of the biotinylated derivative of caffeic acid was not given in Materials and Methods section.

2)I suggest to authors to give recommendation concerning the use of phenols in meat industry, such a limit of the ratio added due to their oxidation with formation of quinones.

Minor corrections:

1)The significance of some abbreviations such as SDS-PAGE or TFA were not given.

2) page 11, line 371 . The capitation of the Table 2 should be corrected.

3) Page 9, Figure 6. The unit of measure of the CA concentration should be given.

4) Page 14,  The quality of Figure 7 should be improved.

Author Response

Dear Reviewer,

Thank you very much for giving us a chance and valuable comments to do a revision for our manuscript, foods-2717001. The comments are very helpful for revising and improving our manuscript. We have carefully checked the comments and revised the manuscript, which we hope to meet your approval now. Our responses in this letter and revisions in the manuscript are marked in red color.

Comments 1: The method of synthesis of the biotinylated derivative of caffeic acid was not given in Materials and Methods section.

Response 1: Thanks for your comment. The method of synthesis of the biotinylated derivative of caffeic acid has been described in our previous published article. In view of this, we have given a brief description of the method in Materials and Methods section. (page 3, line 97-107)

 Comments 2: I suggest to authors to give recommendation concerning the use of phenols in meat industry, such a limit of the ratio added due to their oxidation with formation of quinones.

Response 2: Thank you for your kind comment. We have given recommendation concerning the use of phenols in meat industry. (page 16, line 446-462)

Comments 3: The significance of some abbreviations such as SDS-PAGE or TFA were not given.

Response 3: The significance of abbreviations such as SDS-PAGE (page 1, line 19) and TFA (page 4, line 169) have been given in the revised manuscript.

Comments 4:Page 11, line 371 . The capitation of the Table 2 should be corrected.

Response 4: Corrected. (page 12, line 385)

Comments 5:Page 9, Figure 6. The unit of measure of the CA concentration should be given.

Response 5: Done. (page 9, line 317)

Comments 6:Page 14, The quality of Figure 7 should be improved.

Response 6: Thanks for your comment. The quality of Figure 7 has been improved in the revised manuscript. (page 14, line 387)

We would like to give you our special thanks for your good comments. We have tried our best to improve the manuscript in line according to the suggestions. We appreciate your efforts and hope that the changes will meet with your approval. Looking forward to your kind reply.

Best regards,

Changbo Tang ([email protected])

Reviewer 2 Report

Comments and Suggestions for Authors

In this study, the authors developed and validated a new method based on well know molecular  probe technology to accurately identify phenol-protein adducts in meat protein systems  using BSA as a model protein.The method involves the use of BioC, a biotinylated derivative of CA, which can covalently bind to muscle proteins (MP) in the same manner as CA. It was demonstrated that BioC, a biotinylated derivative of CA, can interact with MP in the same way as CA, forming covalent bonds with specific amino acid residues. By using  LC-MS/MS analysis,  CA-adducted proteins,  CA-adducted peptides, and several kinds of CA-adducted sites were identified. It was revealed that CA prefers to react with the -NH2 group site rather than the -SH group of MP, resulting in various phenol-protein adducts. The method provides a interesting tool for the isolation and identification of phenol-protein adducts in meat protein systems the types and numbers of adducted sites, which revealed the mechanism of covalent binding of phenols to proteins. Should be modified as it but adding that proteins with  polyphenols of high molecular weight such as procyanidins and more complexed tannins the results could be much different. 

Author Response

Dear Reviewer,

Thank you very much for giving us a chance and valuable comments to do a revision for our manuscript, foods-2717001. Our responses in this letter are marked in red color.

Comments 1: In this study, the authors developed and validated a new method based on well know molecular  probe technology to accurately identify phenol-protein adducts in meat protein systems  using BSA as a model protein.The method involves the use of BioC, a biotinylated derivative of CA, which can covalently bind to muscle proteins (MP) in the same manner as CA. It was demonstrated that BioC, a biotinylated derivative of CA, can interact with MP in the same way as CA, forming covalent bonds with specific amino acid residues. By using  LC-MS/MS analysis,  CA-adducted proteins,  CA-adducted peptides, and several kinds of CA-adducted sites were identified. It was revealed that CA prefers to react with the -NH2 group site rather than the -SH group of MP, resulting in various phenol-protein adducts. The method provides a interesting tool for the isolation and identification of phenol-protein adducts in meat protein systems the types and numbers of adducted sites, which revealed the mechanism of covalent binding of phenols to proteins. Should be modified as it but adding that proteins with polyphenols of high molecular weight such as procyanidins and more complexed tannins the results could be much different. 

Response 1: Thank you for your kind comment. Our considerations are as follows: In this study, the covalent interactions between plant polyphenols and meat proteins were investigated. However, only low molecular weight polyphenols were used, which may differ from high molecular weight polyphenols, such as proanthocyanidins and more complex tannins, in their interaction patterns and biological effects. High molecular weight polyphenols might form more stable covalent complexes with proteins. Future investigation is therefore required to take into account the molecular characteristics of polyphenols and the environmental factors that may influence the interactions between polyphenols and proteins, and their effects on meat quality. (page 16, line 454-462)

We would like to give you our special thanks for your good comment.

Best regards,

Changbo Tang ([email protected])

Reviewer 3 Report

Comments and Suggestions for Authors

Please address the following comments;

L74-84: summarize the aim to be matched with the title.

-L 104; specify the model of the incubator, apply this issue with all instruments used in the study.

-L 140: remove the repeated words.

-L 144: define ABST, as well all abbreviations in the first time mentioned.

-L 155: define TFA.

-Under Fig. 4 caption: define ND meaning, as well for all similar cases.

-In Table 1: what is the meaning of gene (/)?

-Provide fig. 7 in high quality.

Comments on the Quality of English Language

Should be improved.

Author Response

Dear Reviewer,

Thank you very much for giving us a chance and valuable comments to do a revision for our manuscript, foods-2717001. The comments are very helpful for revising and improving our manuscript. We have carefully checked the comments and revised the manuscript, which we hope to meet your approval now. Our responses in this letter and revisions in the manuscript are marked in red color.

Comments 1: L74-84: summarize the aim to be matched with the title.

Response 1: Thank you for your kind comment. We have carefully improved this section to summarize the aim to be matched with the title in the revised manuscript. (page 2, line 77-79)

 Comments 2: -L 104; specify the model of the incubator, apply this issue with all instruments used in the study.

Response 2: The model of the incubator is “QB-210; Haimen Kylin-Bell Lab Instruments Co., Ltd., Nantong, China”, which has been added in the revised manuscript (page 3, line 112-113). Meanwhile, we have specified the models of all instruments used in the study.

Comments 3: -L 140: remove the repeated words.

Response 3: Done.

Comments 4: -L 144: define ABST, as well all abbreviations in the first time mentioned.

Response 4: TBST (it's actually TBST in the manuscript) is the abbreviation for “Tris Buffered Saline with Tween-20”, which has been added in the revised manuscript (page 4, line 158), as have other abbreviations in the first time mentioned.

Comments 5: -L 155: define TFA.

Response 5: TFA is the abbreviation for “trifluoroacetic acid”, which has been added in the revised manuscript. (page 4, line 169)

Comments 6: -Under Fig. 4 caption: define ND meaning, as well for all similar cases.

Response 6: The meaning of ND is “non-detected”, which indicated that the intensity of corresponding band is too low to be detected. We have annotated it in the revised manuscript. (page 8, line 295)

Comments 7: -In Table 1: what is the meaning of gene (/)?

Response 7: The meaning of gene (/) is that no gene is found match for pdb 5NOL B Chain B Cardiac muscle alpha actin 1 in the database.

Comments 8: -Provide fig. 7 in high quality.

Response 8: Done. (page 14, line 387) 

Comments 9: The Quality of English Language should be improved.

Response 9: Thank you for your kind comment. We have carefully improved the quality of English language with the help of a colleague fluent in English writing.

We would like to give you our special thanks for your good comments. We have tried our best to improve the manuscript in line according to the suggestions. We appreciate your efforts and hope that the changes will meet with your approval. Looking forward to your kind reply.

Best regards,

Changbo Tang ([email protected])